# Posterior circulation acute stroke prognosis early CT scores in predicting functional outcomes: A meta-analysis

Wei-Zhen Lu[1]⊙, Hui-An Lin[2], Chyi-Huey Bai[3,4,5], Sheng-Feng Lin🔟[3,6,7]⊙*

**1** Department of Emergency Medicine, Taipei Medical University-Shuang Ho Hospital, Taipei, Taiwan,
**2** Department of Emergency Medicine, Taipei Medical University Hospital, Taipei, Taiwan, **3** School of Public
Health, College of Public Health, Taipei Medical University, Taipei, Taiwan, **4** Department of Public Health,
School of Medicine, College of Medicine, Taipei Medical University, Taipei, Taiwan, **5** Nutrition Research
Center, Taipei Medical University Hospital, Taipei, Taiwan, **6** Division of Hospitalist, Department of Internal
Medicine, Far Eastern Memorial Hospital, New Taipei, Taiwan, **7** Department of Neurology, Far Eastern
Memorial Hospital, New Taipei, Taiwan

⊙ These authors contributed equally to this work.
* d508105001@tmu.edu.tw

Posterior circulation acute stroke prognosis early
CT scores in predicting functional outcomes: A
meta-analysis. PLoS ONE 16(2): e0246906. https://
doi.org/10.1371/journal.pone.0246906

Calderón Guardia, CCSS, COSTA RICA

**Data Availability Statement:** Datasets for this
research are published literatures (please referred
to the list of references) and the codes used for
statistical analysis with Stata software, version 15

## Abstract

### Background and purpose

Patients with posterior circulation acute ischemic stroke exhibit varied clinical presentations
and functional outcomes. Whether posterior circulation acute stroke prognosis early com-
puted tomography scores (PC-ASPECTS) predict unfavorable functional outcomes (UFO)
for patients treated with different therapeutic regimens is unclear.

### Methods

According to PRISMA guidelines, we performed a systematic search of electronic data-
bases for studies assessing the functional outcomes of posterior circulation acute ischemic
stroke using baseline PC-ASPECTS. The following three scales of PC-ASPECTS were
retrieved: UFO prediction by using PC-ASPECTS per score decrease, UFO prediction by
using binary PC-ASPECTS with a cut-off value, and the difference in PC-ASPECTS
between patients with unfavorable and favorable functional outcomes. Moreover, a sub-
group analysis was conducted for patients treated with intra-arterial endovascular treatment
(IA-EVT) only. Sensitivity analysis with different definition of UFO and image modalities
were also conducted.

### Results

In total, 25 studies were included. In scale 1, PC-ASPECTS significantly predicted UFO
(odds ratio [OR]: 1.66 per score decrease, 95% confidence interval [CI]: 1.32–2.07). In
scale 2, binary PC-ASPECTS with a cut-off value between 6 and 9 significantly predicted
UFO (OR: 3.91, 95% CI: 2.54–6.01). In scale 3, patients with UFO had lower PC-ASPECTS
than those with favorable outcomes (standardized mean difference [SMD]: −0.67, 95% CI:
−0.8 to −0.55). For patients treated with IA-EVT only, the scales demonstrated consistent

(StataCorp, Texas, USA) was available in the supplemental file.

**Funding:** The author(s) received no specific funding for this work.

**Competing interests:** The authors have declared that no competing interests exist.

**Abbreviations:** ASPECTS, the Alberta stroke program early computed tomography computed tomography score; BAO, Basilar artery occlusion; CI, confidence interval; CT, computed tomography; CTA-SI, computed tomography angiography-source imaging; DWI, diffusion-weighted imaging; MRI, magnetic resonance imaging; mRS, modified Rankin Scale; NIHSS, National Institutes of Health Stroke Scale; OR, odds ratio; PC-ASPECTS, the posterior circulation Alberta stroke program early CT score; SMD, standardized mean difference; UFO, unfavorable functional outcomes.

results. Sensitivity analysis showed PC-ASPECTS significantly predicted UFO in both definitions of modified Rankin Scale $\geq 3$ and $\geq 4$, and magnetic resonance imaging was preferred imaging modality for PC-ASPECTS evaluation.

## Conclusion

Baseline PC-ASPECTS is effective in predicting UFO for patients with posterior circulation acute ischemic stroke treated with different therapeutic regimens.

## Introduction

Posterior circulation infarction accounts for one-fifth to one-fourth of all incidents of acute ischemic stroke [1,2] and has varied clinical presentations [3–5]. Acute basilar artery occlusion (BAO), the most devastating form of posterior circulation infarction, has a mortality rate of approximately 80% [6,7]. Prompt recanalization for BAO and other posterior circulation infarctions was proven to reduce morbidity and mortality [8,9]. Nevertheless, most scoring tools assessing whether stroke patients are candidates for thrombolytic therapy, such as the National Institutes of Health Stroke Scale (NIHSS; which is weighted more for anterior circulation symptoms and signs) [10] and the Alberta stroke program early computed tomography (CT) score (ASPECTS) (assessing the early ischemic changes in the middle cerebral artery territory) [11], are not universally applicable to posterior circulation infarction.

Similar to ASPECTS, posterior circulation Alberta stroke program early CT scores (PC-ASPECTS) allots 10 points to assess early ischemic changes on computed tomography angiography-source imaging (CTA-SI) or diffusion-weighted imaging (DWI) of magnetic resonance imaging (MRI) [12]. These 10 points provide semiquantitative estimates of the region of posterior circulation infarction, which include each side of the cerebellum (1 point), occipital lobe (1 point), thalamus (1 point), pons (2 point), and midbrain (2 point). Unlike anterior circulation stroke with a consensus ASPECTS of <6 as a robust predictor of unfavorable functional outcomes (UFO) and a relative contraindication of endovascular treatment [13,14], studies of posterior circulation infarction have demonstrated a discrepancy when predicting outcomes using baseline PC-ASPECTS. To address this issue, we performed a meta-analysis to determine whether baseline PC-ASPECTS effectively discriminate between unfavorable and favorable outcomes.

## Materials and methods

### Literature search

The first study to propose the scoring method of PC-ASPECTS was published on January 24, 2008. All relevant studies published in English were identified by searching PubMed, Embase, and Scopus from January 1, 2008, to July 31, 2020. The systematic search to identify eligible studies was conducted in accordance with PRISMA guidelines [15]. The keywords and MeSH terms within the search strategy included "stroke (MeSH)," "cerebrovascular disorders (MeSH)," "basilar artery (MeSH)," "vertebrobasilar insufficiency (MeSH)," "endovascular procedures (MeSH)," "thrombolytic therapy (MeSH)," "mechanical thrombolysis (MeSH)," "posterior circulation," "PC-ASPECTS," and their combination (S1 Table). Ethical approval for this study was waived by the Institutional Review Board of Far Eastern Memorial Hospital, New Taipei, Taiwan.

## Study selection and quality assessment

The inclusion criteria were as follows: (1) studies reporting a series of patients with posterior circulation acute ischemic stroke and baseline PC-ASPECTS and (2) studies reporting the functional outcomes in terms of modified Rankin Scale (mRS) on day 90. The exclusion criteria were as follows: (1) abstract papers, (2) case reports, (3) review articles, (4) small studies with a sample size less than 10, (5) studies published in languages other than English, and (5) studies including pediatric populations. Independent screening of titles and abstracts was conducted by two authors (Lu WZ and Lin SF) to identify the studies that met the inclusion criteria. A final list was assembled based on their consensus (Lu WZ and Lin SF). The Newcastle–Ottawa Scale was used for quality assessment (S2 Table). Each retrieved study was critically appraised by two independent reviewers (Lu WZ and Lin SF). Disagreements between them were resolved by other two authors (Lin HA and Bai CH).

## Data collection and outcomes

The study outcome was the prediction of UFO using baseline PC-ASPECTS for patients with posterior circulation acute ischemic stroke receiving any type of treatment. UFO was defined as an mRS of 3–6 or 4–6 on day 90. The association between UFO prediction and baseline PC-ASPECTS for patients was assessed using the following three scales: (1) UFO prediction by using PC-ASPECTS per score decrease, (2) UFO prediction by using binary PC-ASPECTS with a cut-off value, and (3) the difference in baseline PC-ASPECTS between posterior circulation infarction patients with UFO and those with favorable functional outcomes.

## Statistical analysis

We estimated the odds ratio (OR) and standardized mean difference (SMD) and their 95% confidence interval (CI) for each cohort. Heterogeneity for each outcome was assessed using the Higgins index ($I^2$); thereafter, the DerSimonian–Laird random effects model was applied. We defined the Higgins index value ($I^2$) > 50% as substantial heterogeneity [16]. To evaluate heterogeneity and bias, a funnel plot was generated, followed by Begg's test. Analyses were performed using metan package for Stata software, version 15 (StataCorp, Texas, USA). All reported $P$ values were two-sided and were considered as statistically significant at <0.05. A step-down Bonferroni and Hochberg adjustment were used to address the problems of multiple testing.

## Subgroup analyses

We conducted subgroup analyses assessed UFO prediction by using the baseline PC-ASPECTS for patients with intra-arterial endovascular treatment (IA-EVT). We also assessed UFO prediction by using baseline PC-ASPECTS in the above three scales. A random effects meta-regression test was performed to assess the heterogeneity caused by the potential modifiers of the imaging modality of CT or MRI and the definition of UFO based on an mRS of 3–6 or 4–6.

## Sensitivity analyses

We performed two sensitivity analyses: (1) varied definition of UFO by mRS of 3–6 or 4–6, (2) varied imaging modalities of CT and MRI. The UFO Outcomes in three scales were analyzed as well.

## Results

### Search results

The flowchart describes the study selection process (Fig 1). During the initial search, a total of 2488 studies were identified, and only 25 studies met the inclusion and exclusion criteria. Characteristics of the included studies are summarized in Table 1. We included studies with prospective and retrospective cohort designs. In scale 1 (PC-ASPECTS per score decrease), a total of 8 studies with 609 patients were included. In scale 2 (binary PC-ASPECTS by a cut-off value), a total of 7 studies with 844 patients were enrolled. In scale 3 (SMD of PC-ASPECTS between patients with unfavorable and those with favorable functional outcomes), a total of 19 studies with 1186 patients were included in the analysis. For enrolled studies, the quality

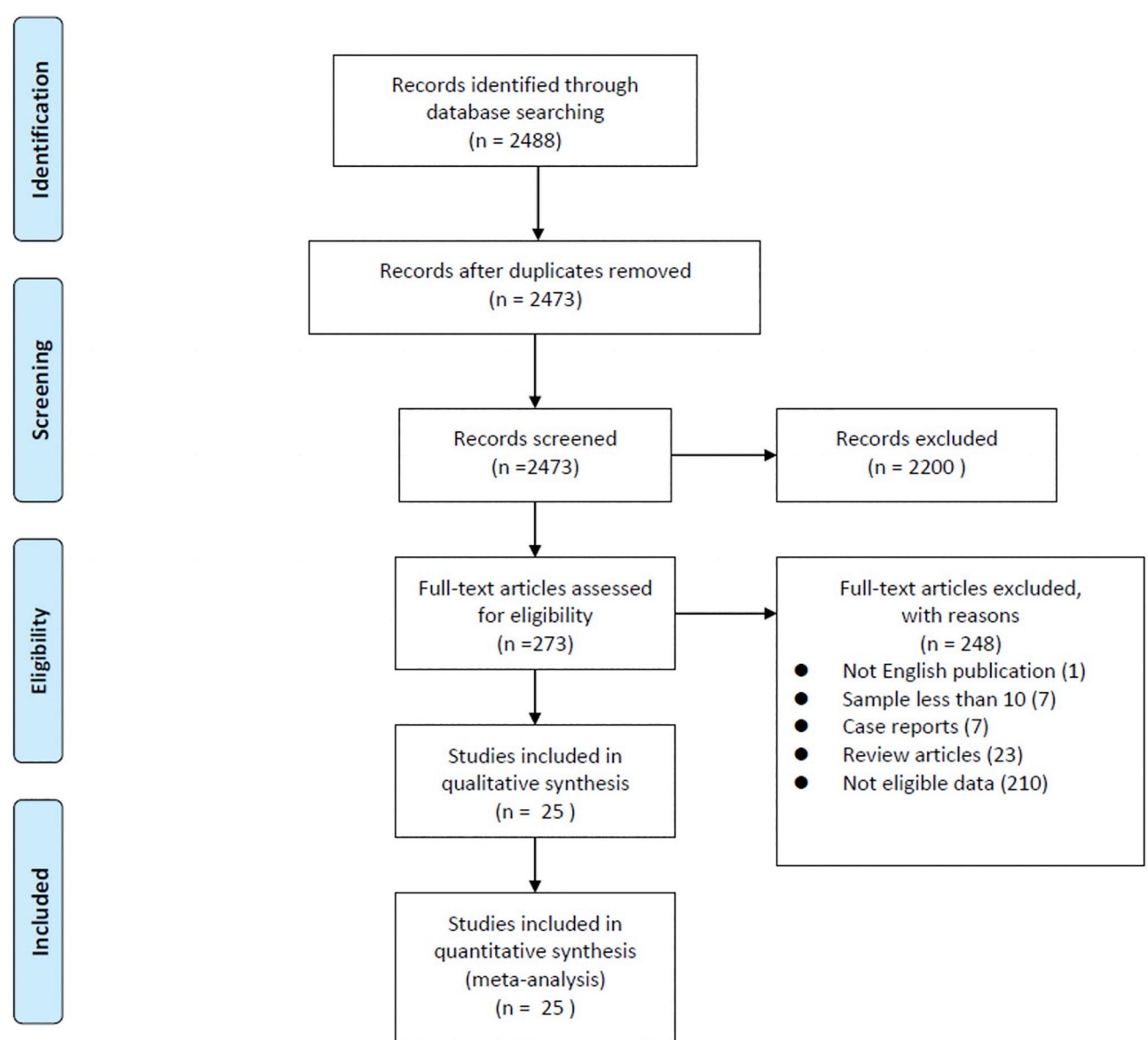

**Fig 1. PRISMA flowchart of search strategy.**

**Table 1. Summary of methodological characteristics of included studies pertinent to the study outcomes.**

| Study | Year | Sample size (number) | PC-ASPECTS cut-off | Clinical Measures | Imaging | Patients treated with thrombolysis (number) |
|---|---|---|---|---|---|---|
| Scale 1: Unfavorable outcome prediction by PC-ASPECTS per score decrease | | | | | | |
| Junji Uno et al. [17] | 2020 | 50 | – | 90-day mRS of 0–2 vs 3–6 | CTP | 50 with IA-EVT |
| Xuelei Zhang et al. [18] | 2019 | 103 | – | 90-day mRS of 0–2 vs 3–6 | MRI (DWI) | 103 with IA-EVT |
| Sheng-Feng Lin et al. [19] | 2018 | 125 | – | 90-day mRS of 0–2 vs 3–6 | MRI (DWI) | 1 with IV thrombolysis only |
| Chuanhui Li et al. [20] | 2018 | 68 | – | 90-day mRS of 0–3 vs 4–6 | CTA-SI or MRI (DWI) | 68 with IA-EVT |
| Woong Yoon et al. [21] | 2015 | 50 | – | 90-day mRS of 0–2 vs 3–6 | MRI (DWI) | 14 with IV thrombolysis, 50 with IA thrombectomy |
| Isabelle Mourand et al. [22] | 2014 | 31 | – | 90-day mRS of 0–2 vs 3–6 | MRI (DWI) | 19 with IA-EVT and IV thrombolysis 12 with IA-EVT only |
| Simon Nagel et al. [23] | 2012 | 50 | – | 90-day mRS of 0–2 vs 3–6 | MRI (DWI) | 41 with recanalization therapy |
| Hideaki Tei et al. [24] | 2010 | 132 | – | 90-day mRS of 0–2 vs 3–6 | MRI (DWI) | (No data provided) |
| Scale 2: Unfavorable outcomes prediction by binary PC-ASPECTS with a cut-off value | | | | | | |
| Volker Maus et al. [25] | 2019 | 104 | 0–8, 9–10 | 90-day mRS of 0–2 vs 3–6 | CTA-SI or MRI (DWI) | 104 with IA EVT 49 with IV + IA thrombolysis |
| Sheng-Feng Lin et al. [19] | 2018 | 125 | 0–7, 8–10 | 90-day mRS of 0–2 vs 3–6 | MRI (DWI) | 1 with IV thrombolysis only |
| Gang Luo et al. [26] | 2018 | 69 | 0–5, 6–10 | 90-day mRS of 0–2 vs 3–6 | MRI (DWI) | 69 with IA EVT |
| Woo-Jin Lee et al. [27] | 2017 | 292 | 0–6, 7–10 | 90-day mRS of 0–2 vs 3–6 | MRI (DWI) | 62 with recanalization therapy |
| Woong Yoon et al. [21] | 2015 | 50 | 0–6, 7–10 | 90-day mRS of 0–2 vs 3–6 | MRI (DWI) | 50 with IA EVT |
| Puetz et al. [28] (BASIC registry) | 2011 | 158 | 0–7, 8–10 | 90-day mRS of 0–2 vs 3–6 | CTA-SI | 15 with IV thrombolysis only 53 with IA EVT only 28 with IV and IA thrombolysis |
| Puetz et al. [12] (BASIC registry) | 2008 | 46 | 0–7, 8–10 | 90-day mRS of 0–3 vs 4–6 | CTA-SI | 36 with any thrombolysis 19 with IV thrombolysis 29 with IA thrombolysis |
| Scale 3: PC-ASPECTS scores between unfavorable and favorable outcomes | | | | | | |
| Beilei Chen et al. [29] | 2019 | 68 | – | 90-day mRS of 0–2 vs 3–6 | CT | 68 with IA-EVT |
| Xuelei Zhang et al. [18] | 2019 | 103 | – | 90-day mRS of 0–2 vs 3–6 | MRI (DWI) | 103 with IA-EVT |
| Fana Alemseged et al. [30] | 2019 | 60 | – | 90-day mRS of 0–3 vs 4–6 | CTA-SI, CTP | 28 with IV thrombolysis, 25 with IA-EVT |
| Francisco Antunes Dias et al. [31] | 2019 | 27 | – | 90-day mRS of 0–3 vs 4–6 | CTA | 27 with IA-EVT |
| Sheng-Feng Lin et al. [19] | 2018 | 125 | – | 90-day mRS of 0–2 vs 3–6 | MRI (DWI) | 1 with IV thrombolysis only |
| Alexandros Rentzos et al. [32] | 2018 | 110 | – | 90-day mRS of 0–2 vs 3–6 | CTA-SI | 110 with IA-EVT |
| Chuanhui Li et al. [20] | 2018 | 68 | – | 90-day mRS of 0–3 vs 4–6 | CTA-SI or MRI (DWI) | 68 with IA-EVT |
| Jun Young Chang et al. [33] | 2017 | 59 | – | 90-day mRS of 0–3 vs 4–6 | MRI (DWI) | 20 with IV thrombolysis 59 with IA-EVT |
| Francisco Antunes Dias et al. [34] | 2017 | 63 | – | 90-day mRS of 0–3 vs 4–6 | NCCT | 15 with IV thrombolysis 19 with IA-EVT 29 with no recanalization |

(*Continued*)

**Table 1.** (Continued)

| Study | Year | Sample size (number) | PC-ASPECTS cut-off | Clinical Measures | Imaging | Patients treated with thrombolysis (number) |
|---|---|---|---|---|---|---|
| Woo-Jin Lee et al. [35] | 2017 | 98 | – | 90-day mRS of 0–2 vs 3–6 | MRI (DWI) | 98 with no recanalization |
| Robert Fahed et al. [36] | 2017 | 34 | – | 90-day mRS of 0–2 vs 3–6 | MRI (DWI) | 34 with IA-EVT |
| Seungnam Son et al. [37] | 2016 | 35 | – | 90-day mRS of 0–2 vs 3–6 | MRI (DWI) | 35 with IA-EVT |
| Valerio Da Ros et al. [38] | 2016 | 15 | – | 90-day mRS of 0–3 vs 4–6 | CTA-SI | 15 with IA-EVT |
| Woong Yoon et al. [21] | 2015 | 50 | – | 90-day mRS of 0–2 vs 3–6 | MRI (DWI) | 14 with IV thrombolysis, 50 with IA thrombectomy |
| S. Mundiyanapurath et al. [39] | 2015 | 22 | – | 90-day mRS of 0–2 vs 3–6 | MRI (DWI, SWI) | 4 with IV thrombolysis only 4 with IA EVT 14 with IV and IA thrombolysis |
| Isabelle Mourand et al. [22] | 2014 | 31 | – | 90-day mRS of 0–2 vs 3–6 | MRI (DWI) | 19 with IA-EVT and IV thrombolysis 12 with IA-EVT only |
| Simon Nagel et al. [23] | 2012 | 50 | – | 90-day mRS of 0–2 vs 3–6 | MRI (DWI) | 41 with recanalization therapy |
| Alexander Karameshev et al. [40] | 2011 | 36 | – | 90-day mRS of 0–3 vs 4–6 | MRI (DWI) | 36 with IA-EVT |
| Hideaki Tei et al. [24] | 2010 | 132 | – | 90-day mRS of 0–2 vs 3–6 | MRI (DWI) | (No data provided) |

BASIC, The Basilar Artery International Cooperative Study; CTP, Computed tomography perfusion scan; CTA-SI, computed tomography angiography-source imaging; ECASS: European Cooperative Acute Stroke Study; EVT, endovascular therapy; mRS, modified Rankin Scale; NCCT, non-contrast computed tomography; IA, intra-arterial; IV, intravenous; MRI, magnetic resonance imaging; SWI, susceptibility-weighted imaging. IV thrombolysis defined as treatment with tissue plasminogen activator.

assessment was evaluated by Newcastle-Ottawa Scale. All of our included studies attained high quality with a total score of $\geq 8$ (See online S2 Table).

## Outcomes: PC-ASPECTS for patients with any treatment

**(1) OR meta-analysis for PC-ASPECTS per score decrease.** This meta-analysis compared the baseline PC-ASPECTS between patients with UFO and those with favorable functional outcomes on day 90. Meta-analysis results were expressed as ORs for UFO prediction by using PC-ASPECTS per score decrease (Fig 2A). All eight studies, except one, individually revealed that the baseline PC-ASPECTS significantly predicted UFO [22]. Overall, the meta-analysis of these studies revealed a significant association between UFO prediction and PC-ASPECTS (OR: 1.66, 95% CI: 1.32–2.07 per score decrease; $I^2 = 59.6\%$, $P = 0.016$). The funnel plot (Fig 2B) followed by Begg's test exhibited no significant publication bias for the eight studies ($P = 0.174$).

**(2) OR meta-analysis for PC-ASPECTS by cut-off score.** UFO prediction by using binary PC-ASPECTS with a cut-off value was analyzed for the included seven studies in this section (Fig 3). These included studies were classified into four subgroups according to their cut-off values. Of these studies, one, two, three, and one had PC-ASPECTS cut-offs of $\leq 6$, $\leq 7$, $\leq 8$, and $\leq 9$, respectively. All the studies individually revealed a significant association between UFO prediction and baseline binary PC-ASPECTS. The subgroup OR meta-analysis demonstrated a significant association between UFO prediction and binary PC-ASPECTS with cut-offs of $\leq 7$ (OR: 4.22, 95% CI: 2.06–8.62; $I^2 = 0.0\%$, $P = 0.808$) and $\leq 8$ (OR: 4.25, 95% CI: 1.5–12.07; $I^2 = 65.6\%$, $P = 0.055$).

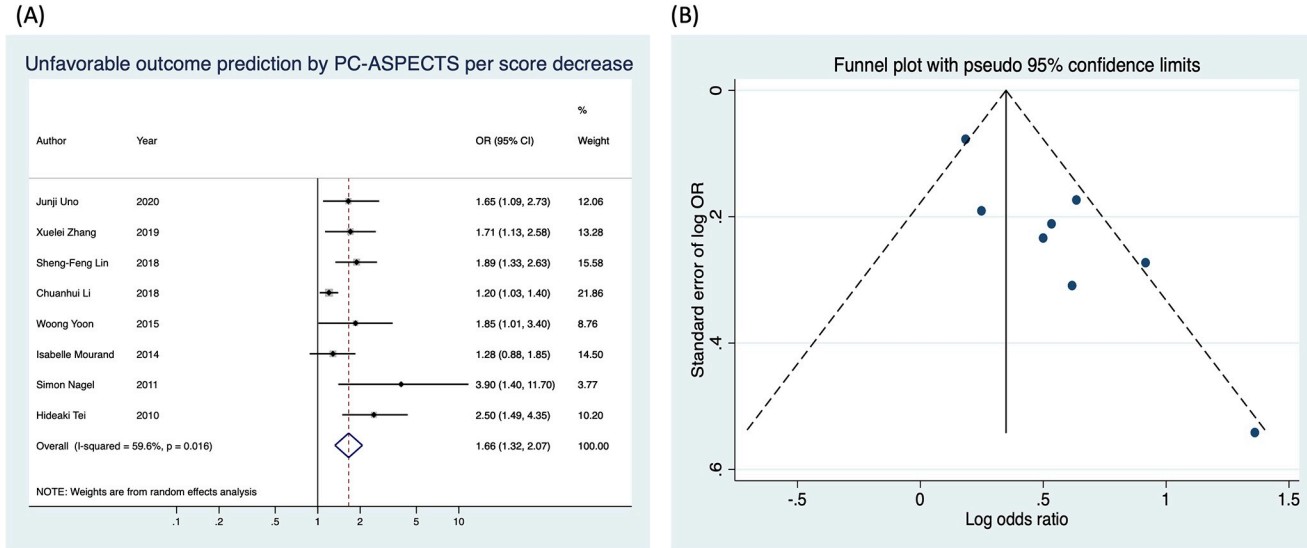

**Fig 2. Unfavorable functional outcome prediction by using PC-ASPECTS per score decrease.** (A) The forest plot. (B) The funnel plot.

Generally, the overall OR meta-analysis consistently displayed a significant association between UFO and binary PC-ASPECTS at any cut-off value, without heterogeneity (OR: 3.91, 95% CI: 2.54–6.01; $I^2 = 17.8\%$, $P = 0.294$). Besides, we performed a test of significance for effect size with by using step-down Bonferroni method and Hochberg adjustment. The adjusted P values by step-down Bonferroni and Hochberg adjustment methods showed PC-ASPECTS of $\leq 7$ with the smallest adjusted P value (S4 Table).

**(3) SMD between unfavorable and favorable outcomes of PC-ASPECTS.** The difference in PC-ASPECTS between unfavorable and favorable functional outcomes was analyzed for studies that provided PC-ASPECTS using a continuous scale (Fig 4A). In total, 19 studies were eligible in this meta-analysis. Of these, 10 studies exhibited a significant difference in score of PC-ASPECTS for patients with unfavorable and favorable outcomes, whereas 9 studies demonstrated no significant difference. Among studies revealing no significant difference, one study [38] paradoxically displayed low PC-ASPECTS in the favorable functional outcome group (SMD: 0.55, 95% CI: −0.5 to 1.61). Generally, the overall SMD meta-analysis result showed significant differences in PC-ASPECTS despite significant heterogeneity (SMD: −0.67, 95% CI: −0.8 to −0.55; $I^2 = 73.4\%$, $P = 0.000$). The funnel plot of 19 studies (Fig 4B) followed by Begg's test revealed no significant publication bias ($P = 0.529$).

## Subgroup analyses: PC-ASPECTS for patients treated with IA-EVT

**(1) OR meta-analysis for PC-ASPECTS per score.** In total, three studies [18,22,23] involving patients treated with IA-EVT were included (Fig 5A). The overall results showed a significant association between UFO prediction and PC-ASPECT per score decrease for patients treated with IA-EVT, without heterogeneity (OR: 1.67, 95% CI: 1.09–2.58; $I^2 = 51.2\%$, $P = 0.129$).

**(2) OR meta-analysis for binary PC-ASPECTS by cut-off score.** Three studies [21,25,26] were included for PC-ASPECTS with cut-off points at 6, 7, and 9 (Fig 5B). The overall results demonstrated a significant association between UFO and binary PC-ASPECTS, with no heterogeneity (OR: 4.91, 95% CI: 2.40–10.05; $I^2 = 0.0\%$, $P = 0.845$).

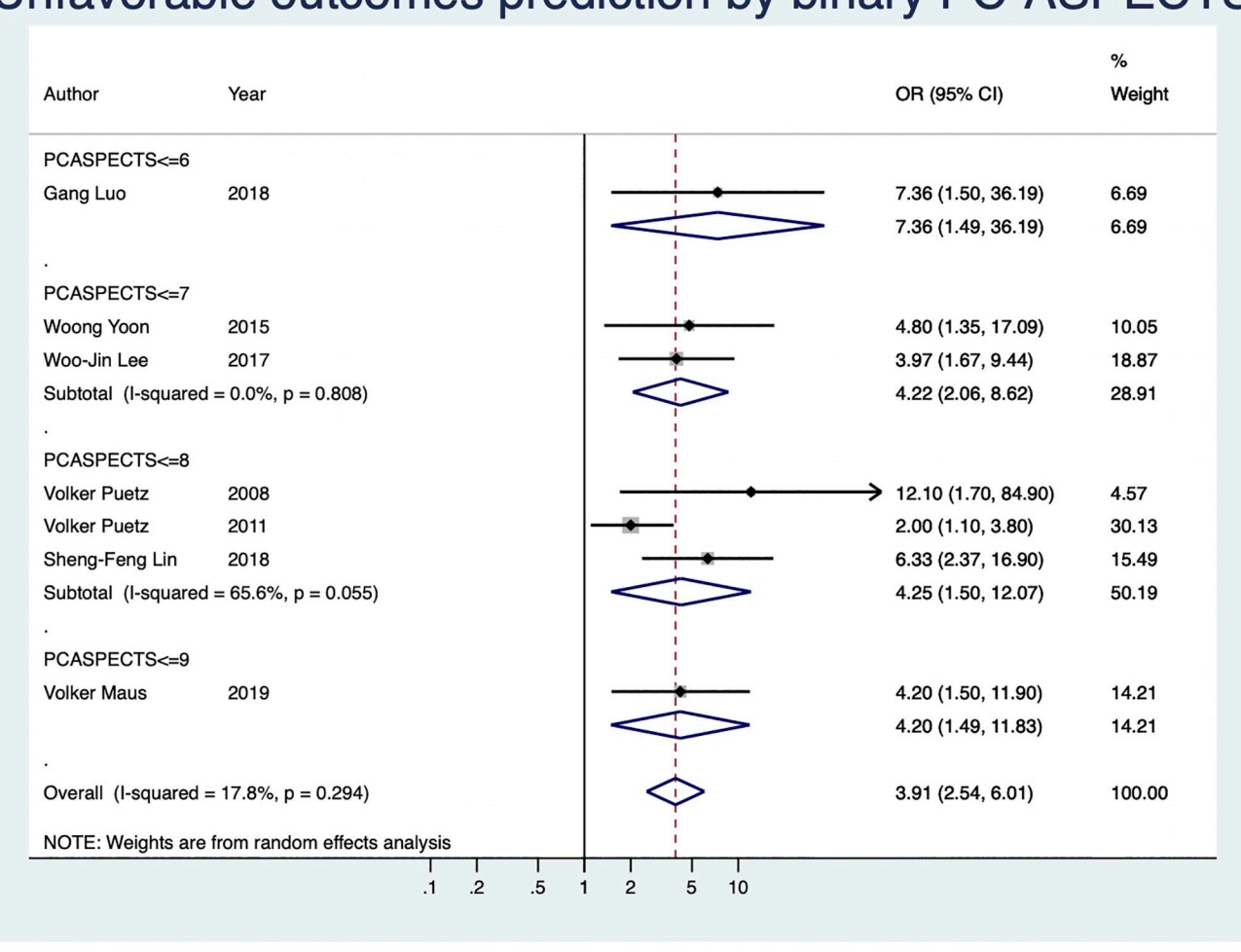

**Fig 3. The forest plot of unfavorable functional outcome prediction by using PC-ASPECTS with a cut-off value.**

**(3) SMD meta-analysis between unfavorable and favorable outcomes of PC-AS-PECTS.** In total, 13 studies were eligible in this meta-analysis (Fig 5C). Of these, six studies showed no significance in scores of PC-ASPECTS for patients with unfavorable and favorable outcomes, whereas seven studies showed no significant difference. Consistent with the analysis for patients on any treatment regimen, the overall SMD meta-analysis result for IA-EVT displayed a significant difference in PC-ASPECTS between unfavorable and favorable outcomes (SMD: −0.54, 95% CI: −0.7 to −0.38; $I^2$ = 78.7%, $P$ = 0.000). The funnel plot (Fig 5D) followed by Begg's test revealed no significant publication bias ($P$ = 0.246). In addition, a meta-regression (S3 Table) exhibited a borderline significant association between the SMD of PC-AS-PECTS and the imaging modality of CT or MRI ($P$ = 0.05), with no significant differences between different definitions of mRS ($P$ = 0.422).

## Sensitivity analyses

We conducted two sensitivity analyses: (1) with varied imaging modalities of CT or MRI, and (2) definition of UFO according to an mRS 3–6 or 4–6.

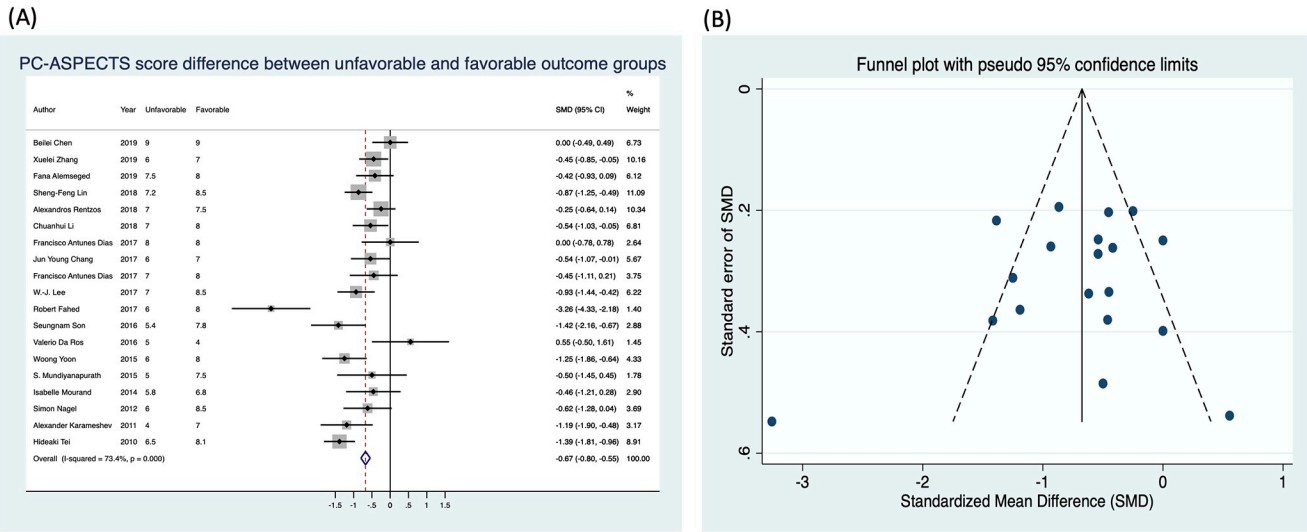

**Fig 4. The standardized mean difference in PC-ASPECTS scores between patients with unfavorable and favorable outcomes.**

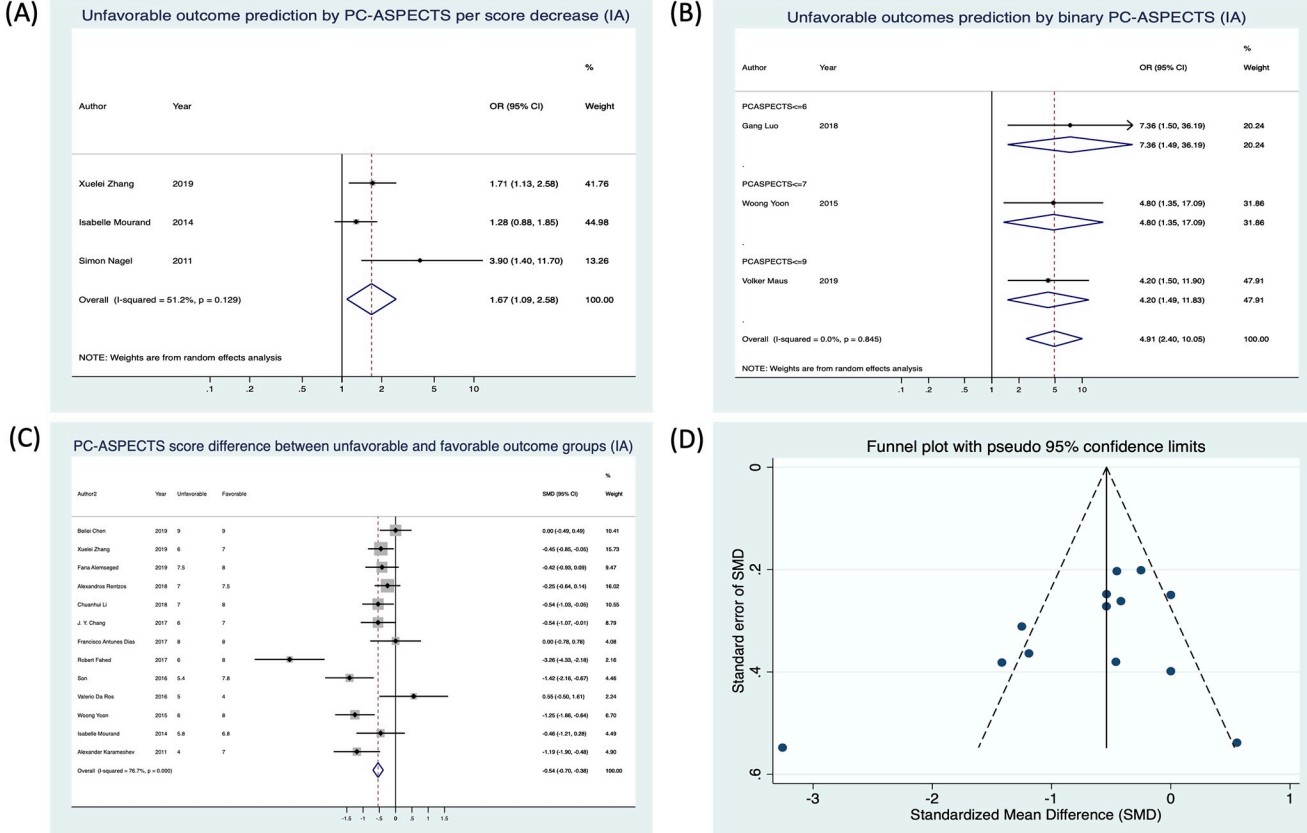

**Fig 5. Unfavorable functional outcome prediction for patients managed by intra-arterial endovascular treatment (subgroup analysis).** (A) Scale by using PC-ASPECTS per score decrease. (B) Scale by using PC-ASPECTS with a cut-off value. (C) Scale by standardized mean difference in PC-ASPECTS scores between patients with unfavorable and favorable outcomes. (D) Funnel plot of scale by standardized mean difference.

**(1) Different imaging modality.** *OR meta-analysis for PC-ASPECTS per score decrease.* Six studies with MRI (OR: 1.81, 95% CI: 1.44–2.27; $I^2$ = 26.3%, P = 0.237) and one study with CT (OR: 1.65, 95% CI: 1.09–2.73) showed scoring of PC-ASPECTS significantly associated with UFO (S1 Fig).

*OR meta-analysis for binary PC-ASPECTS by cut-off score.* Four studies with MRI showed PC-ASPECTS significantly associated with UFO (OR: 5.09, 95% CI: 2.96–8.78; $I^2$ = 0.0%, P = 0.869), whereas two studies with CT revealed no significant association between PC-AP-SECTS and UFO (OR: 3.84, 95% CI: 0.70–20.90; $I^2$ = 66.2%, P = 0.085) (S2 Fig).

*SMD meta-analysis between unfavorable and favorable outcomes of PC-ASPECTS.* Twelve studies with MRI showed PC-ASPECTS significantly predicted UFO (SMD: −0.93, 95% CI: −1.09 to −0.77; $I^2$ = 69.9%, P = 0.000), whereas six studies with CT demonstrated no significant association between PC-ASPECTS and UFO (SMD: −0.19, 95% CI: −0.42 to 0.03; $I^2$ = 0.0%, P = 0.524) (S3 Fig).

**(2) Different definition of unfavorable outcomes by mRS.** With different definitions of UFO, PC-ASPECTS showed significantly predicted the UFO with varied definitions of mRS of 3–6 and 4–6 in three scales (S4, S5 and S6 Figs).

## Discussion

This meta-analysis confirmed that baseline PC-ASPECTS effectively predict functional outcomes for patients with posterior circulation acute ischemic stroke. We analyzed its association with UFO using the following three scales: (1) PC-ASPECTS per score decrease, (2) binary PC-ASPECTS with a cut-off value, and (3) difference in PC-ASPECTS between patients with unfavorable and favorable outcomes. Each decrease in the PC-ASPECTS score was significantly associated with UFO prediction. Although the results among all cut-off value demonstrated significance, we recommended a cut-off value of 7 for PC-ASPECTS effectively discriminated between patients with unfavorable and favorable functional outcomes. Furthermore, for patients treated with IA-EVT, subgroup analyses indicated a robust association between UFO prediction and PC-ASPECTS.

### Different therapeutic regimens

The included studies revealed the following varied thrombolytic therapeutic regimens for patients with posterior circulation acute ischemic stroke: IV thrombolysis, IA-EVT, combination of both, and no recanalization. Subgroup analyses for patients with IA-EVT revealed consistent results with any therapeutic regimen, and the significant heterogeneity disappeared for IA-EVT only in scale 1. This suggested that the heterogeneity in these studies might have been resulted from different intervention regimens. Despite different therapeutic regimens, the results of this meta-analysis still indicated that PC-ASPECTS have a good predicting value for identifying UFO. However, we recommend that further studies should confirm the ability of PC-ASPECTS to predict functional outcomes in subgroups treated with no recanalization or IV thrombolysis because few studies were included in our meta-analysis.

### Studies with different binary PC-ASPECTS cut-offs

The included studies displayed a diverse range of cut-off values of binary PC-ASPECTS for UFO prediction (from PC-ASPECTS of ≤6 to 9). Most studies adopted cut-off points of ≤7 to 8. Despite all cut-off values displaying a significant association between UFO prediction and PC-ASPECTS, we considered that a cut-off of ≤7 would be more appropriate because of a strong effect size. The statistical test for effect size (ES) supported a cut-off point of

PC-ASPECTS ≤ 7 being the best cut-off value due to the smallest raw and adjusted P value (S4 Table).

### Studies with varied definitions of UFO and imaging modalities

In total, 25 studies were included in our meta-analysis. We noted a discrepancy in the definitions of UFO based on the mRS score. Of these, 17 studies defined favorable outcome as mRS scores ≤2, and the rest defined favorable outcome as mRS scores ≤3. In our analyses, we applied each study's individual UFO definition because of a vague boundary between mRS scores 2 and 3 in clinical practice. Our sensitivity analysis suggested PC-ASPECTS was applicable in predicting UFO for studies with a varied boundary of mRS.

Besides, we noted a disparity of imaging modalities of CT and MRI were used for evaluating PC-ASPECTS. The sensitivity analysis suggested MRI should be preferred modality for employing PC-ASPECTS. The association between PC-ASPECTS and UFO with MRI showed significant in three scales, whereas with CT demonstrated significant for OR meta-analysis for PC-ASPECTS per score decrease only.

At last, the meta-regression analysis (S3 Table) for patients treated with IA-EVT supports the negligible effect on heterogeneity by UFO definition (P = 0.412). Conversely, imaging modality of CT or MRI caused a significant effect on the heterogeneity of this study (P = 0.039).

### Limitation

Our study had some limitations. First, high heterogeneity was observed in most of our analyses, which may be caused by study designs, therapeutic regimens, and imaging modalities. Second, none of the included studies involving patients treated with IA-EVT were randomized trials, and data were derived from cohort studies without control groups. However, publication bias was mostly eliminated in our analyses.

### Conclusion

This meta-analysis of retrospective studies revealed that low baseline PC-ASPECTS are significantly associated with UFO. PC-ASPECTS of <7 was the most reasonable cut-offs for predicting UFO. Imaging modality and recanalization regimen were responsible for the heterogeneity in our meta-analysis. Even a one-point decrease in PC-ASPECTS significantly affects the outcome.

### Supporting information

**S1 Checklist.**
(PDF)

**S1 Fig. Unfavorable outcome prediction by PC-ASPECTS per score decrease (varied imaging modality).**
(PDF)

**S2 Fig. Unfavorable outcome prediction with binary PC-ASPECTS (varied imaging modality).**
(PDF)

**S3 Fig. PC-ASPECTS score difference between unfavorable and favorable outcome (varied imaging modality).**
(PDF)

**S4 Fig. Unfavorable outcome prediction by PC-ASPECTS per score decrease (varied definition of mRS).**
(PDF)

**S5 Fig. Unfavorable outcome prediction by binary PC-ASPECTS (varied definition of mRS).**
(PDF)

**S6 Fig. PC-ASPECTS score difference between unfavorable and favorable outcome (varied definition of mRS).**
(PDF)

**S1 Table. Search strategy.**
(PDF)

**S2 Table. Quality measure of included studies by the Newcastle-Ottawa quality assessment scale.**
(PDF)

**S3 Table. Meta-regression of standardized mean difference of PC-ASPECTS between unfavorable and favorable outcomes for patients with intra-arterial endovascular treatment.**
(PDF)

**S4 Table. Significant test for effect size for each cut-off of PC-ASPECTS.**
(PDF)

**S1 File.**
(PDF)

## Acknowledgments

Lin SF conceived the idea; all authors have contributed to the design of the study; Lu WZ, Lin HA, Bai CH, and Lin SF, collected and managed the data, including quality control. Lin SF provided statistical advice on study design and analyzed the data. Lin SF supervised the conduct of the study. Lu WZ and Lin SF drafted the manuscript, and all authors contributed substantially to its revision. Lu WZ and Lin SF contributed equally. Lin SF takes responsibility for the paper as a whole.

## Author Contributions

**Conceptualization:** Wei-Zhen Lu, Sheng-Feng Lin.

**Data curation:** Sheng-Feng Lin.

**Formal analysis:** Sheng-Feng Lin.

**Investigation:** Wei-Zhen Lu, Hui-An Lin, Chyi-Huey Bai.

**Methodology:** Sheng-Feng Lin.

**Supervision:** Sheng-Feng Lin.

**Validation:** Sheng-Feng Lin.

**Writing – original draft:** Wei-Zhen Lu, Sheng-Feng Lin.

**Writing – review & editing:** Sheng-Feng Lin.

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
