## [Decision Letter · Decision Letter 0]

27 Nov 2020

PONE-D-20-33033

Posterior Circulation Acute Stroke Prognosis Early CT Scores in Predicting Functional Outcomes: A Meta-analysis

PLOS ONE

Dear Dr. Lin,

Thank you for submitting your manuscript to PLOS ONE. After careful consideration, we feel that it has merit but does not fully meet PLOS ONE’s publication criteria as it currently stands. Therefore, we invite you to submit a revised version of the manuscript that addresses the points raised during the review process.Psubmit your revised manuscript by Jan 11 2021 11:59PM. If you will need more time than this to complete your revisions, please reply to this message or contact the journal office at plosone@plos.org. Please include the following items when submitting your revised manuscript:

We look forward to receiving your revised manuscript.

Kind regards,

Miguel A. Barboza, MD, MSc

Academic Editor

PLOS ONE

Journal Requirements:

2.In your Data Availability statement, you have not specified where the minimal data set underlying the results described in your manuscript can be found. PLOS defines a study's minimal data set as the underlying data used to reach the conclusions drawn in the manuscript and any additional data required to replicate the reported study findings in their entirety. All PLOS journals require that the minimal data set be made fully available. For more information about our data policy, please see http://journals.plos.org/plosone/s/data-availability.

Reviewers' comments:

Reviewer's Responses to Questions

**Comments to the Author**

1. Is the manuscript technically sound, and do the data support the conclusions?

Reviewer #1: Yes

Reviewer #2: Yes

2. Has the statistical analysis been performed appropriately and rigorously? 

Reviewer #1: No

Reviewer #2: Yes

3. Have the authors made all data underlying the findings in their manuscript fully available?

Reviewer #1: No

Reviewer #2: Yes

4. Is the manuscript presented in an intelligible fashion and written in standard English?

Reviewer #1: Yes

Reviewer #2: Yes

5. Review Comments to the Author

Reviewer #1: This article innovates as it aims to establish if PC-ASPECTS may predict functional outcomes in patients with acute ischemic stroke. This has been an issue with NIHSS and ASPECTS as both scales are focused in anterior circulation strokes. The authors followed a systematic review (SR) approach for searching available articles with a meta-analyses (MA) that used a random effects modeling approach to account for heterogeneity. Even though the main objectives of this research were achieved, the presentation of results fails to be clear to the reader at several points of the article and this should be addressed prior to the approval for publication of this article.

1. The authors do not establish if their SR was previously registered at PROSPERO registry. Currently this is highly expected when attempting a SR in order to later compare the results obtained and the methodological approach performed with the protocol published at the beginning of the investigation. I did a search in the PROSPERO registry and could not find this SR registered, I would like to ask the authors if their review was previously registered and if they followed a previously determined protocol for this.

2. The authors confuse along the whole article between subgroups and sensitivity analysis in regards of their comparison of the results between patients with and without Intra-arterial endovascular treatment (IA-EVT). This by no means should be considered a sensitivity analysis, specially if you are reporting different outcomes obtained after a different intervention was applied. This should be formally analyzed and described as subgroup analysis. On the contrary, evaluating the differential effect of CT/MRI and different definitions of UFO based on mRS, could be better considered as sensitivity analysis as they would be assessing the same outcome but defined differently, yet not assessing the effect of different interventions. I highly recommend the authors to thoroughly review Cochrane Handbook of Systematic Reviews Chapter 10 to understand better the distinction between sensitivity and subgroup analysis, as both terms are commonly confused.

3. In similar terms, the results section should fully describe the results of sensitivity analysis. This was not described in any detail, yet it was referred to in the discussion part under "Studies with varied definitions of UFO and imaging modalities".

4. As I read the article, it becomes very confusing the cut-offs used by the authors for PC-ASPECTS that determined the best effect. In the discussion they chose 7-8 as the best cut-off but there should be more emphasis along the paper, as they describe the results of their analysis, in regards of what cut-offs should be taken as the main predictors, and specifying the effects sizes that lead into determining what cut-offs are best.

5. Some other minor changes that should be made:

- when describing the analysis of only 1 article, reporting 0.0% of heterogeneity (I2) is redundant and misleading. It is rather obvious that one article will not be heterogeneous on its own.

- it would be good for the reader to describe the authors cut-off points for heterogeneity, otherwise their interpretation of the I2 values seem rather arbitrary. I recommend Chapter 10 of Cochran SR Handbook where they establish cute-offs for heterogeneity.

- Please correct every time you refer to this search as "systemic search". This terminology is incorrect and should be "systematic research".

-Last paragraph of the discussion states "However, publication bias was excluded in our analyses". This is a daring phrase as funnel plots cannot fully exclude publication bias, please rephrase this sentence accordingly.

- It is highly expected in the PRISMA chart that the authors specify the reasons that made them exclude papers. Please list reasons for exclusion.

- Finally, I saw that the supplementary tables have a review of quality of the data retrieved yet there is no description of this in the results or discussion. I would like to know as a reader the quality of the data retrieved.

Reviewer #2: This is a very precise, informative and well written article. The predictive value of PC- ASPECTS is a very pertinent subject since the posterior circulation stroke are usually overlooked in the major trials.

I only have one question for the authors. Regarding the following statement

Furthermore, for patients treated with IA-EVT, sensitivity analyses only indicated a robust association between UFO prediction and PC-ASPECTS.

Why do you think there was only robust association between UFO and PC- ASPECTS when treated with AI EVT? Do you think this is due to the heterogenicity of the IA – EVT studies or perhaps that the therapy itself may modify the outcomes.

This is a high quality and pertinent paper; therefore, I recommend its publication.

6. PLOS authors have the option to publish the peer review history of their article (what does this mean?). If published, this will include your full peer review and any attached files.

Reviewer #1: No

Reviewer #2: **Yes: **Beatriz Mendez

---

## [Author Response · Author response to Decision Letter 0]

6 Jan 2021

We thank the reviewers for their constructive comments. We have made revisions to the manuscript to address all the questions and comments raised by the three reviewers. We highlights changes made to the original version by setting the text color to red. Our specific responses to each comment are as follows:

Responses to reviewers #1:

√ This article innovates as it aims to establish if PC-ASPECTS may predict functional outcomes in patients with acute ischemic stroke. This has been an issue with NIHSS and ASPECTS as both scales are focused in anterior circulation strokes. The authors followed a systematic review (SR) approach for searching available articles with a meta-analyses (MA) that used a random effects modeling approach to account for heterogeneity. Even though the main objectives of this research were achieved, the presentation of results fails to be clear to the reader at several points of the article and this should be addressed prior to the approval for publication of this article.

● We are grateful for all of your constructive comments. In the revised manuscript, we addressed the all of the additional points in the following sections.

√ 1. The authors do not establish if their SR was previously registered at PROSPERO registry. Currently this is highly expected when attempting a SR in order to later compare the results obtained and the methodological approach performed with the protocol published at the beginning of the investigation. I did a search in the PROSPERO registry and could not find this SR registered, I would like to ask the authors if their review was previously registered and if they followed a previously determined protocol for this.

● Thank you for the comment. We did not previously register this topic in the PROSPERO registry. However, we previously proposed this study to the research committee in the Far Eastern Memorial Hospital, New Taipei, Taiwan on Sep. 24, 2020. This study was also approved for exemption from the review of the Institutional Review Board, Far Eastern Memorial Hospital, New Taipei, Taiwan. The certificate of proposal on Sep. 24, 2020. was attached in the following page.

● Besides, all of the studies enrolled in our analysis were observational retrospective cohort studies. Neither randomized-controlled trial nor interventional study was included in our analysis. The analytic outcomes should not be changed despite no previous registry in the PROSPERO system. However, we were sincerely grateful for the reviewer’s kind reminder for PROSPERO registry to avoid repetitive topic.

● The certificate for proposal in the research committee in the Far Eastern Memorial Hospital, New Taipei, Taiwan (Chinese language). 

√ 2. The authors confuse along the whole article between subgroups and sensitivity analysis in regards of their comparison of the results between patients with and without Intra-arterial endovascular treatment (IA-EVT). This by no means should be considered a sensitivity analysis, specially if you are reporting different outcomes obtained after a different intervention was applied. This should be formally analyzed and described as subgroup analysis. On the contrary, evaluating the differential effect of CT/MRI and different definitions of UFO based on mRS, could be better considered as sensitivity analysis as they would be assessing the same outcome but defined differently, yet not assessing the effect of different interventions. I highly recommend the authors to thoroughly review Cochrane Handbook of Systematic Reviews Chapter 10 to understand better the distinction between sensitivity and subgroup analysis, as both terms are commonly confused.

● Thank you for the comment. We thoroughly reviewed Cochrane Handbook of Systematic Reviews Chapter 10 for distinction between sensitivity analysis and subgroup analysis. The description for outcomes of IA-EVT treatment was revised as “subgroup analysis” in the manuscript. (Please see line 173-179, methods section; line 263-297, results section; line 333 discussion section; Fig 5A-5D)

√ 3. In similar terms, the results section should fully describe the results of sensitivity analysis. This was not described in any detail, yet it was referred to in the discussion part under "Studies with varied definitions of UFO and imaging modalities".

● Thank you for the comment. We added the additional analysis for studies with varied definitions of UFO and imaging modalities. 

● Our result showed MRI was preferred for PC-ASPECTS evaluation. Sensitivity analysis showed significant association between UFO and PC-ASPECTS assessed by MRI in all three scales, whereas no significant association was found between UFO and PC-ASPECTS by CT in scales of binary PC-ASPECTS and SMD. 

● Besides, both the definition of UFO by mRS 3-6 and 4-6 both showed lower PC-ASPECTS significantly predicted UFO in all three scales. 

(Please see line 181-184, methods section; line 299-321, results section; line 359-378, discussion section; S1-S6 Fig)

√ 4. As I read the article, it becomes very confusing the cut-offs used by the authors for PC-ASPECTS that determined the best effect. In the discussion they chose 7-8 as the best cut-off but there should be more emphasis along the paper, as they describe the results of their analysis, in regards of what cut-offs should be taken as the main predictors, and specifying the effects sizes that lead into determining what cut-offs are best.

● Thank you for the comment. We revised this section by providing the test results for effect size (ES) for each cut-off for PC-ASPECTS. In the following figure, the test result for ES by STATA 15 software was shown. On considering the z score and raw P value, we thought PC-ASPECTS of ≤ 7 should be the best cut-off.

● Besides, we performed an adjustment for P value by using step-down Bonferroni method and Hochberg adjustment to address the problems of multiple testing. The adjusted P values by step-down Bonferroni and Hochberg adjustment methods also supported PC-ASPECTS of ≤ 7 being the best cut-off value due to the smallest adjusted P value. We summarized our findings in the following table and added this information to S4 Table. (Please see line 237-243, results section; line 355-357, discussion section; S4 Table)

√ 5. Some other minor changes that should be made:

- when describing the analysis of only 1 article, reporting 0.0% of heterogeneity (I2) is redundant and misleading. It is rather obvious that one article will not be heterogeneous on its own.

● Thank you for the comment. We revised our data reporting in Fig. 3 and Fig. 5. 

√ - it would be good for the reader to describe the authors cut-off points for heterogeneity, otherwise their interpretation of the I2 values seem rather arbitrary. I recommend Chapter 10 of Cochran SR Handbook where they establish cute-offs for heterogeneity.

● Thank you for the comment. We referred to the Cochrane Handbook of Systematic Reviews Chapter 10. We also cited a reference for our objective judgement for heterogeneity. We defined the Higgins index value (I2) > 50% as substantial heterogeneity. (Please see line 165-166, methods section)

√ - Please correct every time you refer to this search as "systemic search". This terminology is incorrect and should be "systematic research".

● Thank you for the comment. We have corrected our terminology. 

(Please see line 59, abstract section; line 130, methods section)

√ - Last paragraph of the discussion states "However, publication bias was excluded in our analyses". This is a daring phrase as funnel plots cannot fully exclude publication bias, please rephrase this sentence accordingly.

● Thank you for kind comment. We have revised our phrase for this sentence. 

(Please see line 385, limitation section)

√ - It is highly expected in the PRISMA chart that the authors specify the reasons that made them exclude papers. Please list reasons for exclusion.

● Thank you for the comment. We have added the reasons for excluding papers.

(Please see Fig 1 PRISMA chart)

√ - Finally, I saw that the supplementary tables have a review of quality of the data retrieved yet there is no description of this in the results or discussion. I would like to know as a reader the quality of the data retrieved.

● Thank you for the comment. We added the description for a review of quality of the data.

(Please see line 146-149, methods section; line 196-198, results section)

Responses to reviewers #2:

√ This is a very precise, informative and well written article. The predictive value of PC- ASPECTS is a very pertinent subject since the posterior circulation stroke are usually overlooked in the major trials. I only have one question for the authors. Regarding the following statement:

Furthermore, for patients treated with IA-EVT, sensitivity analyses only indicated a robust association between UFO prediction and PC-ASPECTS. Why do you think there was only robust association between UFO and PC- ASPECTS when treated with AI EVT? Do you think this is due to the heterogenicity of the IA – EVT studies or perhaps that the therapy itself may modify the outcomes.

● Thank you for the comment. 

● Our main analysis supported scoring of PC-ASPECTS was universally applicable for posterior circulation acute ischemic stroke with varied therapeutic regimens.

● In the literature reviews, we noted studies performing IA-EVT sharing higher conformance in their treatment plan. The subgroup analysis of IA-EVT was performed since previous studies reported that thrombectomy or stent-retriever therapy may greatly improve the functional outcomes for acute ischemic stroke than conservative medical therapy.

● In the subgroup analysis, we proved that PC-ASPECTS was still applicable for patients treated IA-EVT. Thus, the robustness of PC-ASPECTS in predicting unfavorable functional outcome was evident. However, this should not be interpreted as non-applicability of PC-ASPECTS for stroke patients treated with IV thrombolysis or conservative medical therapy. 

(Please see line 336-348, discussion section)

---

## [Decision Letter · Decision Letter 1]

28 Jan 2021

Posterior Circulation Acute Stroke Prognosis Early CT Scores in Predicting Functional Outcomes: A Meta-analysis

PONE-D-20-33033R1

Dear Dr. Lin,

We’re pleased to inform you that your manuscript has been judged scientifically suitable for publication and will be formally accepted for publication once it meets all outstanding technical requirements.

Kind regards,

Miguel A. Barboza, MD, MSc

Academic Editor

PLOS ONE

Additional Editor Comments (optional):

Reviewers' comments:

Reviewer's Responses to Questions

**Comments to the Author**

1. If the authors have adequately addressed your comments raised in a previous round of review and you feel that this manuscript is now acceptable for publication, you may indicate that here to bypass the “Comments to the Author” section, enter your conflict of interest statement in the “Confidential to Editor” section, and submit your "Accept" recommendation.

Reviewer #1: All comments have been addressed

Reviewer #2: All comments have been addressed

2. Is the manuscript technically sound, and do the data support the conclusions?

Reviewer #1: Yes

Reviewer #2: Yes

3. Has the statistical analysis been performed appropriately and rigorously? 

Reviewer #1: Yes

Reviewer #2: Yes

4. Have the authors made all data underlying the findings in their manuscript fully available?

Reviewer #1: Yes

Reviewer #2: Yes

5. Is the manuscript presented in an intelligible fashion and written in standard English?

Reviewer #1: Yes

Reviewer #2: Yes

6. Review Comments to the Author

Reviewer #1: Authors fully addressed all my concerns and the current manuscript reads better. They were able to describe better the statistic approach that led to their conclusions. To my appraisal this article is more understandable now as it was before, for the common reader that might not now about systematic reviews.

Reviewer #2: (No Response)

7. PLOS authors have the option to publish the peer review history of their article (what does this mean?). If published, this will include your full peer review and any attached files.

Reviewer #1: No

Reviewer #2: **Yes: **Beatriz Mendez Gonzalez

---

## [Editor Report · Acceptance letter]

4 Feb 2021

PONE-D-20-33033R1 

Posterior Circulation Acute Stroke Prognosis Early CT Scores in Predicting Functional Outcomes: A Meta-analysis 

Dear Dr. Lin:

I'm pleased to inform you that your manuscript has been deemed suitable for publication in PLOS ONE. Congratulations! Your manuscript is now with our production department. 

Kind regards, 

on behalf of

Dr. Miguel A. Barboza 

Academic Editor

PLOS ONE